# Leveraging Grammar and Reinforcement Learning for Neural Program Synthesis

**Rudy Bunel**[*]
University of Oxford
rudy@robots.ox.ac.uk

**Matthew Hausknecht**
Microsoft Research
matthew.hausknecht@microsoft.com

**Jacob Devlin**[*]
Google
jacobdevlin@google.com

**Rishabh Singh**
Microsoft Research
risin@microsoft.com

**Pushmeet Kohli**[*]
Deepmind
pushmeet@google.com

## Abstract

Program synthesis is the task of automatically generating a program consistent with a specification. Recent years have seen proposal of a number of neural approaches for program synthesis, many of which adopt a sequence generation paradigm similar to neural machine translation, in which sequence-to-sequence models are trained to maximize the likelihood of known reference programs. While achieving impressive results, this strategy has two key limitations. First, it ignores *Program Aliasing*: the fact that many different programs may satisfy a given specification (especially with incomplete specifications such as a few input-output examples). By maximizing the likelihood of only a single reference program, it penalizes many *semantically* correct programs, which can adversely affect the synthesizer performance. Second, this strategy overlooks the fact that programs have a strict syntax that can be efficiently checked. To address the first limitation, we perform reinforcement learning on top of a supervised model with an objective that explicitly maximizes the likelihood of generating semantically correct programs. For addressing the second limitation, we introduce a training procedure that directly maximizes the probability of generating syntactically correct programs that fulfill the specification. We show that our contributions lead to improved accuracy of the models, especially in cases where the training data is limited.

## 1 Introduction

The task of program synthesis is to automatically generate a program that is consistent with a specification such as a set of input-output examples, and has been studied since the early days of Artificial Intelligence (Waldinger and Lee, 1969). There has been a lot of recent progress made on *neural program induction*, where novel neural architectures inspired from computation modules such as RAM, stack, CPU, turing machines, and GPU (Graves et al., 2014; Joulin and Mikolov, 2015; Kurach et al., 2016; Graves et al., 2016; Reed and de Freitas, 2016; Kaiser and Sutskever, 2016) have been proposed to train these architectures in an end-to-end fashion to mimic the behavior of the desired program. While these approaches have achieved impressive results, they do not return explicit interpretable programs, tend not to generalize well on inputs of arbitrary length, and require a lot of examples and computation for learning each program. To mitigate some of these limitations, *neural program synthesis* approaches (Johnson et al., 2017; Parisotto et al., 2017; Devlin et al., 2017b) have been recently proposed that learn explicit programs in a Domain-specific language (DSL) from as few as five input-output examples. These approaches, instead of using a large number of input-output examples to learn a single program, learn a large number of different programs, each from just a few input-output examples. During training, the correct program is provided as reference, but at test time, the learnt model generates the program from only the input-output examples.

While neural program synthesis techniques improve over program induction techniques in certain domains, they suffer from two key limitations. First, these approaches use supervised learning

---
[*]Work performed at Microsoft Research

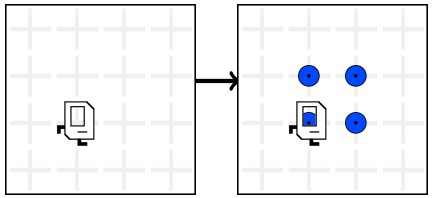

(a) Partial Specification as IO Pair



**Program A**

```
def run():
  repeat(4):
    putMarker()
    move()
    turnLeft()
```

**Program B**

```
def run():
  while(noMarkersPresent):
    putMarker()
    move()
    turnLeft()
```



Figure 1: **Program Aliasing** is one difficulty of program synthesis: For the input-output specification given in (1a), both programs are *semantically correct*. However, supervised training would penalize the prediction of Program B, if A is the ground truth.

with reference programs and suffer from the problem of *Program Aliasing*: For a small number of input-output examples, there can be many programs that correctly transform inputs to outputs. The problem is the discrepancy between the single supervised reference program and the multitude of correct programs. Figure 1 shows an example of this: if maximizing the probability of ground truth program, predicting Program B would be assigned a high loss even though the two programs are semantically equivalent for the input-output example. Maximum likelihood training forces the model to learn to predict ground truth programs, which is different from the true objective of program synthesis: predicting *any* consistent program. To address this problem, we alter the optimization objective: instead of maximum likelihood, we use policy gradient reinforcement learning to directly encourage generation of *any* program that is consistent with the given examples.

The second limitation of neural program synthesis techniques based on sequence generation paradigm (Devlin et al., 2017b) is that they often overlook the fact that programs have a strict syntax, which can be checked efficiently. Similarly to the work of Parisotto et al. (2017), we explore a method for leveraging the syntax of the programming language in order to aggressively prune the exponentially large search space of possible programs. In particular, not all sequences of tokens are valid programs and syntactically incorrect programs can be efficiently ignored both during training and at test time. A syntax checker is an additional form of supervision that may not always be present. To address this limitation, we introduce a neural architecture that retains the benefits of aggressive syntax pruning, even without assuming access to the definition of the grammar made in previous work (Parisotto et al., 2017). This model is jointly conditioned on syntactic and program correctness, and can implicitly learn the syntax of the language while training.

We demonstrate the efficacy of our approach by developing a neural program synthesis system for the Karel programming language (Pattis, 1981), an educational programming language, consiting of control flow constructs such as loops and conditionals, making it more complex than the domains tackled by previous neural program synthesis works.

This paper makes the following key contributions:

- We show that Reinforcement Learning can directly optimize for generating any consistent program and improves performance compared to pure supervised learning.
- We introduce a method for pruning the space of possible programs using a syntax checker and show that explicit syntax checking helps generate better programs.
- In the absence of a syntax checker, we introduce a model that jointly learns syntax and the production of correct programs. We demonstrate this model improves performance in instances with limited training data.

## 2 RELATED WORK

Program synthesis is one of the fundamental problems in Artificial Intelligence. To the best of our knowledge, it can be traced back to the work of Waldinger and Lee (1969) where a theorem prover was used to construct LISP programs based on a formal specification of the input-output relation. As formal specification is often as complex as writing the original program, many techniques were developed to achieve the same goal with simpler partial specifications in the form of input-output (IO) examples (Amarel, 1970; Summers, 1977). Rule-based synthesis approaches have recently been

successful in delivering on the promise of Programming By Example (Lieberman, 2001), the most widely known example being the FlashFill system (Gulwani et al., 2012) in Excel. However, such systems are extremely complicated to extend and need significant development time from domain experts to provide the pruning rules for efficient search.

As a result, the use of Machine Learning methods have been proposed, based on Bayesian probabilistic models (Liang et al., 2010) or Inductive Logic programming (Muggleton, 1991; Muggleton et al., 2014) to automatically generate programs based on examples. Recently, inspired by the success of Neural Networks in other applications such as vision (Krizhevsky et al., 2012) or speech recognition (Graves et al., 2013) differentiable controllers were made to learn the behaviour of programs by using gradient descent over differentiable version of traditional programming concepts such as memory addressing (Graves et al., 2014), manipulating stacks (Joulin and Mikolov, 2015; Grefenstette et al., 2015), register machines (Kurach et al., 2016), and data manipulation (Neelakantan et al., 2016). These approaches to program induction however tend to struggle with generalization, especially when presented with inputs of a different dimension than the one they were trained with and require a very large amount of training data. Some exceptions to this include Neural Programmer Interpreters (Reed and de Freitas, 2016) and its extensions (Li et al., 2017; Cai et al., 2017) that learn from program traces rather than only examples. However, they still learn a different model for each program and are computationally expensive, unlike our system that uses a single model for learning a large number of programs.

A series of recent works aim to infer explicit program source code with the assumption that code structure provides an inductive bias for better generalization. In particular, explicitly modeling control flow statements such as conditionals and loops can often lead to programs capable of generalizing, regardless of input size (Gaunt et al., 2016; Bunel et al., 2016; Riedel et al., 2017). One remaining drawback of these approaches is the need to restart learning from scratch for each new program. Thus they are largely unsuited for the situation of synthesizing a new program on-the-fly from very few examples. The latest developments use large datasets of artificially generated programs and learn to map embeddings of IO examples to information about the programs to generate. Balog et al. (2017) produce scores over attributes, to be used as heuristics to speed up search-based techniques. Parisotto et al. (2017) use their dataset to learn probability over the expansion rules of a predefined grammar, while (Devlin et al., 2017b) directly predict the source code of the programs. These last two methods use supervised training to maximize the likelihood of a single reference program, while we directly optimize for the generation of any consistent program.

Our approach to optimize program correctness is similar in spirit to advances in Neural Machine Translation (Wu et al., 2016; Ranzato et al., 2016) that leverage reinforcement learning to optimize directly for evaluation metrics. Taking advantage of the fact that programs can be syntactically checked and unit tested against the specification examples, we show how to improve on those REINFORCE (Williams, 1992) based methods. Recently, Guu et al. (2017) proposed a method similar to ours based on Maximum Marginal Likelihood to generate programs based on a description in natural language. From an application point of view, our target domain is more complex as our DSL includes control flow operations such as conditionals and loops. Moreover, natural language utterances fully describe the steps that the program needs to take, while learning from IO examples requires planning over potentially long executions. Their approach is more akin to inferring a formal specification based on a natural language description, as opposed to our generation of imperative programs.

Incorporating knowledge of the grammar of the target domain to enforce syntactical correctness has already proven useful to model arithmetic expressions, molecules (Kusner et al., 2017), and programs (Parisotto et al., 2017; Yin and Neubig, 2017). These approaches define the model over the production rules of the grammar; we instead operate directly over the terminals of the grammar. This allows us to learn the grammar jointly with the model in the case where no formal grammar specification is available. Our approach is extremely general and can be applied to the very recently proposed methods for inferring and executing programs for visual reasoning (Johnson et al., 2017) that to the best of our knowledge does not directly explictly encourage grammar consistency of the resulting program.

# 3 PROBLEM OVERVIEW

Before describing our proposed methods, we establish the necessary notation, present the problem setting and describe the general paradigm of our approach.

## 3.1 PROGRAM SYNTHESIS FORMULATION

To avoid confusion with probabilities, we will use the letter $\lambda$ to denote programs. $I$ and $O$ will be used to denote respectively input states and output states and we will use the shortcut $IO$ to denote a pair of corresponding input/output examples. A state constitutes what the programs are going to be operating on, depending on the application domain. In FlashFill-type applications (Parisotto et al., 2017; Devlin et al., 2017b), Input and Output states would be strings of characters, while in our Karel environment, states are grids describing the presence of objects. If we were to apply our method to actual programming languages, states would represent the content of the machine's registers and the memory.

At training time, we assume to have access to $N$ training samples, each training sample consisting of a set of $K$ Input/Output states and a program implementing the mapping correctly:

$$\mathcal{D} = \left\{ \left( \left\{ IO_i^k \right\}_{k=1..K}, \lambda_i \right) \right\}_{i=1..N} \quad \text{such that:} \quad \lambda_i(I_i^k) = O_i^k \quad \forall i \in 1..N, \quad \forall k \in 1..K \quad (1)$$

where $\lambda_i(I_i^k)$ denotes the resulting state of applying the program $\lambda_i$ to the input state $I_i^k$. Our goal is to learn a synthesizer $\sigma$ that, given a set of input/output examples produces a program:

$$\sigma : \quad \left\{ IO^k \right\}_{k=1..K} \quad \longrightarrow \quad \hat{\lambda} \quad (2)$$

We evaluate the programs on a set of test cases for which we have both specification examples and held-out examples:

$$\mathcal{D}_{\text{test}} = \left\{ \left( \left\{ IO_j^{k_{\text{spec}}} \right\}_{k_{\text{spec}}=1..K}, \quad \left\{ IO_j^{k_{\text{test}}} \right\}_{k_{\text{test}}=K+1..K'} \right) \right\}_{j=1..N_{\text{test}}} \quad (3)$$

At test time, we evaluate the performance of our learned synthesizer by generating, for each sample in the test set, a program $\hat{\lambda}_j$. The metric we care about is *Generalization*:

$$\left\{ j \in \{1..N_{\text{test}}\} \text{ such that } \hat{\lambda}_j(I_j^k) == O_j^k \quad \forall k \in \{1..K'\} \right\} \text{ where } \hat{\lambda}_j = \sigma \left( \left\{ IO_j^{k_{\text{spec}}} \right\}_{k_{\text{spec}}=1..K} \right). \quad (4)$$

## 3.2 NEURAL PROGRAM SYNTHESIS ARCHITECTURE

Similar to Devlin et al. (2017b) we use a sequential LSTM-based (Hochreiter and Schmidhuber, 1997) language model, conditioned on an embedding of the input-output pairs. Each pair is encoded independently by a convolutional neural network (CNN) to generate a joint embedding. A succinct description of the architecture can be found in section 6.1 and the exact dimensions are available in the supplementary materials.

Each program is represented by a sequence of tokens $\lambda = [s_1, s_2, ..., s_L]$ where each token comes from an alphabet $\Sigma$. We model the program one token at a time using an LSTM. At each timestep, the input consists of the concatenation of the embedding of the IO pair and of the last predicted token. One such decoder LSTM is run for each of the IO pairs, all using the same weights. The probability of the next token is defined as the Softmax of a linear layer over the max-pooled hidden state of all the decoder LSTMs. A schema representing this architecture can be seen in Figure 2.

The form of the model that we are learning is:

$$p_\theta(\lambda_i \mid \left\{ IO_i^k \right\}_{k=1..K}) = \prod_{t=1}^{L_i} p_\theta(s_t \mid s_1, ..., s_{t-1}, \left\{ IO_i^k \right\}_{k=1..K}) \quad (5)$$

At test time, the most likely programs are obtained by running a beam search. One of the advantages of program synthesis is the ability to execute hypothesized programs. Through execution, we remove syntactically incorrect programs and programs that are not consistent with the observed examples. Among the remaining programs, we return the most likely according to the model.

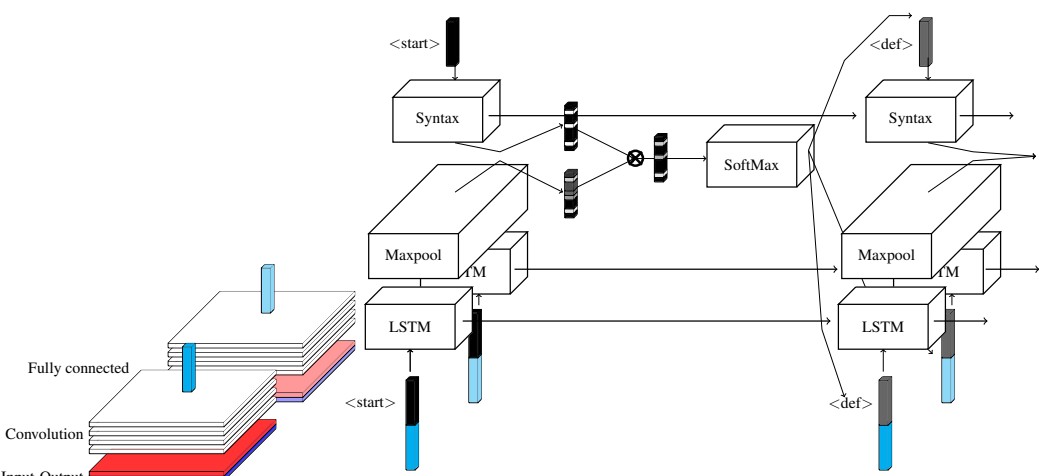

Figure 2: Architecture of our model. Each pair of Input-Output is embedded jointly by a CNN. One decoder LSTM is run for each example, getting fed in a concatenation of the previous token and the IO pair embedding (constant across timestep). Results of all the decoders are maxpooled and the prediction is modulated by the mask generated by the syntax model. The probability over the next token is then obtained by a Softmax transformation.

## 4 OBJECTIVE FUNCTIONS

### 4.1 MAXIMUM LIKELIHOOD OPTIMIZATION

To estimate the parameters $\theta$ of our model, the default solution is to perform supervised training, framing the problem as Maximum Likelihood estimation. Devlin et al. (2017b) follow this approach and use stochastic gradient descent to solve:

$$\theta^\star = \operatorname*{argmax}_\theta \prod_{i=1..N} p_\theta(\lambda_i \mid \{IO_i^i\}_{i=1..K}) = \operatorname*{argmax}_\theta \sum_{i=1..N} \log\left(p_\theta(\lambda_i \mid \{IO_i^k\}_{k=1..K})\right) \quad (6)$$

However, this training objective exhibits several drawbacks. First, at training time, the model is only exposed to the training data distribution, while at test time, it is fed back the token from its own previous predictions. This discrepancy in distribution of the inputs is well known in Natural Language Processing under the name of *exposure bias* (Ranzato et al., 2016).

Moreover, this loss does not represent the true objective of program synthesis. In practice, any equivalent program should be as valid a prediction as the reference one. This property, that we call *program aliasing*, is not taken into account by the MLE training. Ideally, we would like the model to learn to reason about the necessary steps needed to map the input to the output. As a result, the loss shouldn't penalize correct programs, even if they do not correspond to the ground truth.

### 4.2 OPTIMIZING EXPECTED CORRECTNESS

The first modification that we propose is to change the target objective to bring it more in line with the goal of program synthesis. We replace the optimization problem of (6) by

$$\theta^\star = \operatorname*{argmax}_\theta \mathcal{L}_R(\theta), \qquad \text{where } \mathcal{L}_R(\theta) = \sum_{i..N} \left(\sum_\lambda p_\theta(\lambda \mid \{IO_i^k\}_{k=1..K}) \, R_i(\lambda)\right), \quad (7)$$

where $R_i(\lambda)$ is a reward function designed to encode the quality of the sampled programs. Note that this formulation is extremely generic and would allow to represent a wide range of different objective functions. If we assume that we have access to a simulator to run our programs, we can design $R_i$ so as to optimize for generalization on held-out examples, preventing the model to overfit on its inputs. Additional property such as program conciseness, or runtime efficiency could also be encoded into the reward.

Figure 3: Approximation using a beamsearch. All possibles next tokens are tried for each candidates, the $S$ (here 3) most likely according to $p_\theta$ are kept. When an End-Of-Sequence token (green) is reached, the candidate is held out. At the end, the most likely complete sequences are used to construct an approximate distribution, through rescaling.

However, this expressiveness comes at a cost: the inner sum in (7) is over all possible programs and therefore is not tractable to compute. The standard method consists of approximating the objective by defining a Monte Carlo estimate of the expected reward, using $S$ samples from the model. To perform optimization, an estimator of the gradient of the expected reward is built based on the REINFORCE trick (Williams, 1992).

$$\mathcal{L}_R(\theta) \approx \sum_{i=1..N} \sum_{r=1}^{S} \frac{1}{S} R_i(\lambda_r), \qquad \text{where } \lambda_r \sim p_\theta\left(\,\cdot\,\middle|\,\left\{IO_i^k\right\}_{k=1..K}\right)$$

$$\nabla_\theta \mathcal{L}_R(\theta) \approx \sum_{i=1..N} \sum_{r=1}^{S} \frac{1}{S} R_i(\lambda_r) \log\left(p_\theta\left(\,\lambda_r\,\middle|\,\left\{IO_i^k\right\}_{k=1..K}\right)\right)$$

(8)

However, given that we sample from a unique model, there is a high chance that we will sample the same programs repeatedly when estimating the gradient. This is especially true when the model has been pre-trained in a supervised manner. A different approach is to approximate the distribution of the learned distribution by another one with a smaller support.

To obtain this smaller distribution, one possible solution is to employ the $S$ most likely samples as returned by a Beam Search. We generate the embedding of the IO grids and perform decoding, keeping at each step the $S$ most likely candidates prefixes based on the probability $p_\theta$ given by the model. At step $t$, we evaluate $p_\theta(s_1 \ldots s_t, \left\{IO_i^k\right\}_{k=1..K})$ for all the possible next token $s_t$ and all the candidates $(s_1 \ldots s_{t-1})$ previously obtained. The $S$ most likely sequences will be the candidates at the $(t+1)$ step. Figure 3 represents this process.

Based on the final samples obtained, we define a probability distribution to use as an approximation of $p_\theta$ in (7). As opposed to (8), this approximation introduces a bias. It however has the advantage of aligning the training procedure more closely with the testing procedure where only likely samples are going to be decoded. Formally, this corresponds to performing the following approximation of the objective function, (BS$(p_\theta, S)$ being the $S$ samples returned by a beam search with beam size $S$):

$$\mathcal{L}_R(\theta) \approx \sum_{i..N} \left( \sum_{\lambda} q_\theta(\lambda \,|\, \left\{IO_i^k\right\}_{k=1..K}) \, R_i(\lambda) \right)$$

$$\text{where} \qquad q_\theta(\lambda_r \,|\, \left\{IO_i^k\right\}_{k=1..K}) = \begin{cases} \dfrac{p_\theta(\lambda_r \,|\,\left\{IO_i^k\right\}_{k=1..K})}{\sum_{\lambda_r \in \text{BS}(p_\theta, S)} p_\theta(\lambda_r \,|\,\left\{IO_i^k\right\}_{k=1..K})} & \text{if } \lambda_r \in \text{BS}(p_\theta, S) \\ 0 & \text{otherwise.} \end{cases}$$

(9)

With this approximation, the support of the distribution $q_\theta$ is much smaller as it contains only the $S$ elements returned by the beam search. As a result, the sum become tractable and no estimator are needed. We can simply differentiate this new objective function to obtain the gradients of the loss with regards to $p_\theta$ and use the chain-rule to subsequently obtain gradient with regards to $\theta$ necessary for the optimization. Note that if $S$ is large enough to cover all possible programs, we recover the objective of (7).

Based on this more tractable distribution, we can define more complex objective functions. In program synthesis, we have the possibility to prune out several predictions by using the specification. Therefore, we can choose to go beyond optimizing the expected reward when sampling a single

program and optimize the expected reward when sampling a bag of $C$ programs and keeping the best one. This results in a new objective function:

$$\theta^\star = \underset{\theta}{\arg\max} \sum_{i=1..N} \left( \sum_{\{\lambda_1,..,\lambda_C\}\in \text{BS}(p_{\theta,s})^C} \left[ \max_{j\in 1..C} R_i(\lambda_j) \right] \left( \prod_{r\in 1..C} q_\theta \left( \lambda_r \mid \left\{ IO_i^k \right\}_{k=1..K} \right) \right) \right), \quad (10)$$

where $q_\theta$ is defined as previously. We argue that by optimizing this objective function, the model gets the capability of "hedging its bets" and assigning probability mass to several candidates programs, resulting in a higher diversity of outputs.

In the special case where the reward function only takes values in $\{0, 1\}$, as it is when we are using correctness as a reward, this can be more easily computed as:

$$\theta^\star = \underset{\theta}{\arg\max} \sum_{i=1..N} \left( 1 - \left( \sum_{\lambda_r\in\text{BS}(p_\theta,S)} [R_i(\lambda_r) == 0]\, q_\theta(\lambda_r \mid \left\{ IO_i^k \right\}_{k=1..K}) \right)^C \right) \quad (11)$$

The derivation leading to this formulation as well as a description on how to efficiently compute the more general loss(10) can be found in appendix A. Note that although this formulation brings the training objective function closer to the testing procedure, it is still not ideal. It indeed makes the assumption that if we have a correct program in our bag of samples, we can identify it, ignoring the fact that it is possible to have some incorrect program consistent with the IO pairs partial specification (and therefore not prunable). In addition, this represents a probability where the $C$ programs are sampled independently, which in practice we wouldn't do.

## 5 MODEL

### 5.1 CONDITIONING ON THE SYNTAX

One aspect of the program synthesis task is that syntactically incorrect programs can be trivially identified and pruned before making a prediction. As a result, if we use `stx` to denote the event that the sampled program is syntactically correct, what we care about modeling correctly is $p(\lambda \mid \left\{ IO_i^k \right\}_{k=1..K}, \text{stx})$. Using Bayes rule, we can rewrite this:

$$
\begin{aligned}
p\left( \lambda \mid \left\{ IO_i^k \right\}_{k=1..K}, \text{stx} \right) &\propto p\left( \text{stx} \mid \left\{ IO_i^k \right\}_{k=1..K}, \lambda \right) \times p\left( \lambda \mid \left\{ IO_i^k \right\}_{k=1..K} \right) \\
&\propto p\left( \text{stx} \mid \lambda \right) \times p\left( \lambda \mid \left\{ IO_i^k \right\}_{k=1..K} \right)
\end{aligned} \quad (12)
$$

We drop the conditional dependency on the IO pairs in the second line of (12) as the syntactical correctness of the program is independent from the specification when conditioned on the program. We can do the same operation at the token level, denoting by $\text{stx}_{1...t}$ the event that the sequence of the first $t$ tokens $s_1 \cdots s_t$ doesn't contain any syntax error and may therefore be a prefix to a valid program.

$$
\begin{aligned}
p\left( s_t \mid s_1 \cdots s_{t-1}, \left\{ IO_i^k \right\}_{k=1..K}, \text{stx}_{1...t} \right) \propto \quad & p\left( \text{stx}_{1...t} \mid s_1 \cdots s_t \right) \times \\
& p\left( s_t \mid s_1 \cdots s_{t-1}, \left\{ IO_i^k \right\}_{k=1..K} \right)
\end{aligned} \quad (13)
$$

Given a grammar, it is possible to construct a checker to determine valid prefixes of programs. Example applications include compilers to signal syntax errors to the user and autocomplete features of Integrated Development Environments (IDEs) to restrict the list of suggested completions.

The quantity $p\left( \text{stx}_{1...t} \mid s_1 \cdots s_t \right)$ is therefore not a probability but can be implemented as a deterministic process for a given target programming language. In practice, this is implemented by getting at each timestep a mask $M = \{-\inf, 0\}^{|\Sigma|}$ where $M_j = -\inf$ if the $j$-th token in the alphabet is not a valid token in the current context, and 0 otherwise. This mask is added to the output of the network, just before the Softmax operation that normalizes the output to a probability over the tokens.

Conditioning on the syntactical correctness of the programs provides several advantages: First, sampled programs become syntactically correct by construction. At test time, it allows the beam search to only explore useful candidates. It also ensures that when we are optimizing for correctness, the samples used to approximate the distribution are all going to be valid candidates. Restricting the dimension of the space on which our model is defined also makes the problem simpler to learn.

## 5.2 Jointly Learned Syntax

It may not always be feasible to assume access to a syntax checker. In general, we wish to retain the syntax checker's ability to aggressively prune the search in program space without requiring access to the syntax itself. To this end, we propose to represent the syntax checker as a neural network module and learn it jointly. Similar to the base model, we implement learned syntax checking using an LSTM $g_\phi$. Comparing to the decoder LSTM, there are two major differences:

- The syntaxLSTM is conditioned only on the program tokens, not on the IO pairs. This ensures that the learned checker models only the syntax of the language.
- The output of the syntaxLSTM is passed through an elementwise $x \mapsto -\exp(x)$ activation function and added to the decoder LSTM's output. Similar to the mask in Section 5.1, the exponential activation function allows the syntaxLSTM to output high penalties to any tokens deemed syntactically incorrect.

The addition of the syntaxLSTM doesn't necessitate any change to the training procedure as it is simply equivalent to a change of architecture. However, in the supervised setting, when we have access to syntactically correct programs, we have the possibility of adding an additional term to the loss (6) to prevent the model from masking valid programs:

$$\mathcal{L}_{\texttt{syntax}} = -\sum_{i=1\cdots N}\sum_{t=1\cdots L} g_\phi\left(s_t^i \,|\, s_1^i \cdots s_{t-1}^i\right), \qquad \text{where } \lambda_i = [s_1^i, s_2^i \cdots s_L^i] \qquad (14)$$

This loss penalizes the syntaxLSTM for giving negative scores to each token belonging to a known valid program. We use the reference programs as example of valid programs when we perform supervised training.

## 6 Experiments

### 6.1 The Domain: Karel

The Karel programming language is an educational programming language (Pattis, 1981), used for example in Stanford CS introductory classes (cs1) or in the Hour of Code initiative (hoc). It features an agent inside a gridworld (See Figure 1), capable of moving (`move`, `turn{Left, Right}`), modifying world state (`{pick, put}Marker`), and querying the state of the nearby environment for its own markers (`markerPresent`, `noMarkerPresent`) or for natural obstacles (`frontIsClear`, `leftIsClear`, `rightIsClear`). Our goal is to learn to generate a program in the Karel DSL given a small set of input and output grids. The language supports for loops, while loops, and conditionals, but no variable assignment. Compared to the original Karel language, we only removed the possibility of defining subroutines. The specification for the DSL can be found in appendix B.

To evaluate our method, we follow the standard practice (Devlin et al., 2017b; Parisotto et al., 2017; Neelakantan et al., 2016; Balog et al., 2017) and use a synthetic dataset generated by randomly sampling programs from the DSL. We perform a few simple heuristic checks to ensure generated programs have observable effect on the world and prune out programs performing spurious actions (e.g. executing a `turnLeft` just after a `turnRight` for example). For each program, we a set of IO pairs are generated by sampling random input grids and executing the program on them to obtain the corresponding output grids. A large number of them are sampled and 6 are kept for each program, ensuring that all conditionals in a program are hit by at least one of the examples. The first 5 samples serve as the specification, and the sixth one is kept as held-out test pair. 5000 programs are not used for training, and get split out between a validation set and a test set.

We represent the input and output elements as grids where each cell in the grid is a vector with 16 channels, indicating the presence of elements (`AgentFacingNorth`, `AgentFacingSouth`, $\cdots$, `Obstacle`, `OneMarkerPresent`, `TwoMarkersPresent`, $\cdots$). The input and output grids are initially passed through independent convolution layers, before being concatenated and passed through two convolutional residual blocks and a fully connected layer, mapping them to a final 512-dimensional representation. We run one decoder per IO pair and perform a maxpooling operation over the output of all the decoders, out of which we perform the prediction of the next token. Our models are implemented using the Pytorch framework (pyt). Code and data will be made available.

| Top-1 | Full Dataset | | Small Dataset | |
|---|---|---|---|---|
| | Generalization | Exact Match | Generalization | Exact Match |
| **MLE** | 71.91 | **39.94** | 12.58 | 8.93 |
| **RL** | 68.39 | 34.74 | 0 | 0 |
| **RL_beam** | 75.72 | 8.21 | **25.28** | **17.63** |
| **RL_beam_div** | 76.20 | 31.25 | 23.72 | 16.31 |
| **RL_beam_div_opt** | **77.12** | 32.17 | 24.24 | 16.63 |

Table 1: **RL_beam** optimization of program correctness results in consistent improvements in top-1 generalization accuracy over supervised learning **MLE**, even though the exact match of recovering the reference program drops. The improved objective function results in further improvements.

## 6.2 RESULTS

We trained a variety of models on the full Karel dataset containing 1-million examples as well as a reduced dataset containing only $10,000$ examples. In general, the small dataset serves to help understand the data efficiency of the program synthesis methods and is motivated by the expected difficulty of obtaining many labeled examples in real-world program synthesis domains.

The Karel DSL was previously used by Devlin et al. (2017a) to study the relative perfomances of a range of methods depending on the available amount of data. The task considered was however different as they attempted to perform program induction as opposed to program synthesis. Rather than predicting a program implementing the desired transformation, they simply output a specification of the changes that would result from applying the program so a direct number comparison wouldn't be meaningful.

Models are grouped according to training objectives. As a baseline, we use **MLE**, which corresponds to the maximum likelihood objective (Eq.6), similar to the method proposed by Devlin et al. (2017b). Unless otherwise specified, the reward considered for our other methods is generalization: +1 if the program matches all samples, including the held out one and 0 otherwise. **RL** uses the expected reward objective (Eq.7), using REINFORCE to obtain a gradient estimate (Eq.8). **RL_beam** attempts to solve the proxy problem described by Equation (9) and **RL_beam_div** the richer loss function of Equation (10). **RL_beam_div_opt** also optimizes the loss of equation (10) but the reward additionally includes a term inversly proportional to the number of timesteps it takes for the program to finish executing. All RL models are initialized from pretrained supervised models.

**Optimizing for correctness (RL):** Results in Table 1 show that optimizing for the expected program correctness consistently provides improvements in top-1 generalization accuracy. **Top-1 Generalization Accuracy** (Eq. 4) denotes the accuracy of the most likely program synthesized by beam search decoding having the correct behaviour across all input-output examples. We didn't perform any pruning of programs that were incorrect on the 5 specification examples. The improved performance of RL methods confirms our hypothesis that better loss functions can effectively combat the program aliasing problem.

On the full dataset, when optimizing for correctness, **Exact Match Accuracy** decreases, indicating that the RL models no longer prioritize generating programs that exactly match the references. On the small dataset, **RL_beam** methods improves both exact match accuracy and generalization.

Comparing RL_beam to standard RL, we note improvements across all levels of generalization. By better aligning the RL objective with the sampling that happens during beam search decoding, consistent improvements can be made in accuracy. Further improvements are made by encouraging diversity in the beam of solutions (RL_beam_div) and penalizing long running programs (RL_beam_div_opt).

In the settings where little training data is available, RL methods show more dramatic improvements over MLE, indicating that data efficiency of program synthesis methods can be greatly improved by using a small number of samples first for supervised training and again for Reinforcement Learning.

As a side note, we were unable to achieve high performance when training the RL methods from scratch. The necessity of extensive supervised pretraining to get benefits from Reinforcement

| Generalization | Top-1 | Top-5 | Top-50 |
|---|---|---|---|
| **MLE** | 71.91 | 79.56 | **86.37** |
| **RL_beam** | 75.72 | 79.29 | 83.49 |
| **RL_beam_div** | 76.20 | 82.09 | 85.86 |
| **RL_beam_div_opt** | **77.12** | **82.17** | 85.38 |

| Top-1 Generalization | Full Dataset | Small Dataset |
|---|---|---|
| **MLE** | 71.91 | 12.58 |
| **MLE_learned** | 69.37 | **17.02** |
| **MLE_handwritten** | 72.07 | 9.81 |
| **MLE_large** | **73.67** | 13.14 |

Table 2: **Top-k accuracies:** MLE shows greater relative accuracy increases as k increases than RL. Methods employing beam search and diversity objectives reduce this accuracy gap by encouraging diversity in the beam of partial programs.

Table 3: **Grammar prunes the space of possible programs**: On the full dataset, handwritten syntax checking MLE_handwritten improves accuracy over no grammar MLE, although MLE_large shows that simply adding more parameters results in even greater gains. On the small dataset, learning the syntax MLE_learned outperforms handwritten grammar and larger models.

Learning fine-tuning is well-known in the Neural Machine Translation literature (Ranzato et al., 2016; Wu et al., 2016; Wiseman and Rush, 2016; Bahdanau et al., 2017).

Table 2 examines the top-1, top-5, and top-50 generalization accuracy of supervised and RL models. RL_beam methods performs best for top-1 but their advantage drops for higher-rank accuracy. Inspection of generated programs shows that the top predictions all become diverse variations of the same program, up to addition/removal of no-operations (`turnLeft` followed by `turnRight`, full circle obtained by a series of four `turnLeft`). The RL_beam_div objective helps alleviate this effect as does RL_beam_div_opt, which penalizes redundant programs. This is important as in the task of Program Synthesis, we may not necessarily need to return the most likely output if it can be pruned by our specification.

**Impact of syntax:** We also compare models according to the use of syntax: **MLE_handwritten** denotes the use of a handwritten syntax checker (Sec 5.1), **MLE_learned** denotes a learned syntax (Sec 5.2), while no suffix denotes no syntax usage. Table 3 compares syntax models.

On the full dataset, leveraging the handwritten syntax leads to marginally better accuracies than learning the syntax or using no syntax. Given access to enough data, the network seems to be capable of learning to model the syntax using the sheer volume of training examples.

On the other hand, when the amount of training data is limited, learning the syntax produces significantly better performance. By incorporating syntactic structure in the model architecture and objective, more leverage is gained from small training data. Interestingly, the learned syntax model even outperforms the handwritten syntax model. We posit the syntaxLSTM is free to learn a richer syntax. For example, the syntaxLSTM could learn to model the distribution of programs and discourage the prediction of not only syntactically incorrect programs, but also the unlikely ones.

To control for the extra parameters introduced by the syntaxLSTM, we compare against MLE_large, which uses no syntax but features a larger decoder LSTM, resulting in the same number of parameters as MLE_learned. Results show that the larger number of parameters is not enough to explain the difference in performance, which again indicates the utility of jointly learning syntax.

**Analysis of learned syntax:** Section 5.2 claimed that by decomposing our models into two separate decoders, we could decompose the learning so that one decoder would specialize in picking the likely tokens given the IO pairs, while the other would enforce the grammar of the language. We now provide experimental evidence that this decomposition happens in practice.

Table 4 shows the percentage of syntactically correct programs among the most likely predictions of the **MLE + learned** model trained on the full dataset. Both columns correspond to the same set of parameters but the second column doesn't apply the syntaxLSTM's mask to constrain the decoding process. The precipitous drop in syntax accuracy indicates the extent to which the program decoder has learned to rely on the syntaxLSTM to produce syntactically correct programs.

Figure 3 compares the syntax masks generated by the learned and handwritten syntax models while decoding Program A in Figure 1. (3a) shows output of the handwritten syntax checker; (3b) shows

syntaxLSTM output. White cells indicates tokens that are labeled syntactically correct at each decoding step.

Figure 3c analyzes the difference between the handwritten and learned syntax masks. White indicates similar output, which occupies the majority of the visualization. Blue cells correspond to instances where the syntaxLSTM labeled a token correct when it actually was syntactically incorrect. This type of error can be recovered if the program decoder predicts those tokens as unlikely.

On the other hand, red cells indicate the syntaxLSTM predicted a valid token is syntactically incorrect. This type of error is more dangerous because the program decoder cannot recover the valid token once it is declared incorrect. The majority of red errors correspond to tokens which are rarely observed in the training dataset, indicating that the syntaxLSTM learns to model more than just syntax - it also captures the distribution over programs. Given a large enough dataset of real programs, the syntaxLSTM learns to consider non-sensical and unlikely programs as syntactically incorrect, ensuring that generated programs are both syntactically correct and likely.

| % Synctactically Correct | Joint Model | Without Learned Syntax |
|---|---|---|
| Amongst Top1 | 100 % | 0 % |
| Amongst Top5 | 100 % | 0 % |
| Amongst Top50 | 100 % | 0 % |
| Amongst Top100 | 99.79 % | .04 % |

Table 4: Importance of Syntax

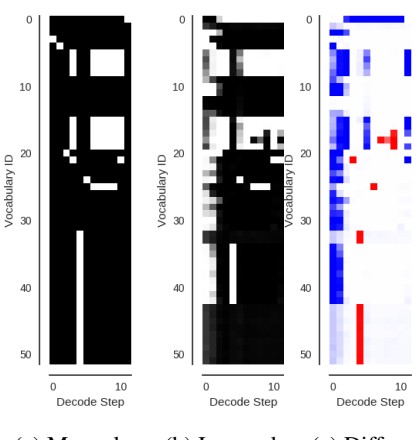

(a) Manual     (b) Learned     (c) Diff

Figure 3: Syntax Comparison

## 7 CONCLUSION

We presented two novel contributions to improve state-of-the-art neural program synthesis techniques. Our first contribution uses Reinforcement Learning to optimize for generating any consistent program, which we show helps in improving generalization accuracy of the learned programs for large training datasets. Our second contribution incorporates syntax checking as an additional conditioning mechanism for pruning the space of programs during decoding. We show that incorporating syntax leads to significant improvements with limited training datasets.

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

## A  COMPUTING THE RICHER LOSS FUNCTION

In this section, we describe how the objective function of Equation (10) can be computed.

We have a distribution $q_\theta$ over $S$ programs, as obtained by performing a beam search over $p_\theta$ and renormalizing. We are going to get $C$ independent samples from this distribution and obtain a reward corresponding to the best performing of all of them.

In the case where the reward function $R_i(\lambda_r)$ is built on a boolean property, such as for example "correctness of the generated program", Equation (10) can be simplified. As $R_i(\lambda_r)$ can only take the values 0 or 1, the term $\max_{j \in 1..C} R_i(\lambda_j)$ is going to be equal to 0 only if all of the $C$ sampled programs give a reward of zero. For each sample, there is a probability of $q_{\text{incorrect}} = \sum_{\lambda_r \in \text{BS}(p_\theta, S)} [R_i(\lambda_r) == 0] \, q_\theta(\lambda_r \mid \{IO_i^k\}_{k=1..K})$ of sampling a program with rewards zero. The probability of not sampling a single correct program out of the $C$ samples is $q_{\text{incorrect}}^C$. From this, we can derive the form of Equation (11). Note that this can be computed without any additional sampling steps as we have a close form solution for this expectation.

In the general case, a similar derivation can be obtained. Assume that the programs outputted by the beam search have associated rewards $R_0, R_1, ..., R_S$ and assume without loss of generality that $R_0 < R_1 < .. < R_S$. The probability of sampling a program with a reward smaller than $R_i$ is $q_{\leq R_i} = \sum_{\lambda_r \in \text{BS}(p_\theta, S)} [R_i(\lambda_r) \leq R_i] \, q_\theta(\lambda_r \mid \{IO_i^k\}_{k=1..K})$ so the probability of obtaining a final reward of less than $R_i$ when sampling C samples is $q_{\leq R_i}^C$. As a result, the probability of obtaining a reward of exactly $R_i$ is $\left( q_{\leq R_i}^C - q_{\leq R_{i-1}}^C \right)$. Based on this, it is easy to evaluate (and differentiate) the loss function described by Equation (10).

## B  KAREL LANGAGE SPECIFICATION

$$
\begin{aligned}
\text{Prog } p \quad &:= \quad \texttt{def run()} : s \\
\text{Stmt } s \quad &:= \quad \texttt{while}(b) : s \mid \texttt{repeat}(r) : s \mid s_1; s_2 \mid a \\
&\quad \mid \quad \texttt{if}(b) : s \mid \texttt{ifelse}(b) : s_1 \texttt{ else} : s_2 \\
\text{Cond } b \quad &:= \quad \texttt{frontIsClear()} \mid \texttt{leftIsClear()} \mid \texttt{rightIsClear()} \\
&\quad \mid \quad \texttt{markersPresent()} \mid \texttt{noMarkersPresent()} \mid \texttt{not } b \\
\text{Action } a \quad &:= \quad \texttt{move()} \mid \texttt{turnRight()} \mid \texttt{turnLeft()} \\
&\quad \mid \quad \texttt{pickMarker()} \mid \texttt{putMarker()} \\
\text{Cste } r \quad &:= \quad 0 \mid 1 \mid \cdots \mid 19
\end{aligned}
$$

Figure 4: The Domain-specific language for Karel programs.

## C  EXPERIMENTS HYPERPARAMETERS

The state of the grid word are represented as a $16 \times 18 \times 18$ tensor. For each cell of the grid, the 16-dimensional vector corresponds to the feature indicated in Table 5

The decoders are two-layer LSTM with a hidden size of 256. Tokens of the DSL are embedded to a 256 dimensional vector. The input of the LSTM is therefore of dimension 768 (256 dimensional of the token + 512 dimensional of the IO pair embedding). The LSTM used to model the syntax is similarly sized but doesn't take the embedding of the IO pairs as input so its input size is only 256.

One decoder LSTM is run on the embedding of each IO pair. The topmost activation are passed through a MaxPooling operation to obtain a 256 dimensional vector representing all pairs. This is passed through a linear layer to obtain a score for each of the 52 possible tokens. We add the output of the model and of the eventual syntax model, whether learned or handwritten and pass it through a SoftMax layer to obtain a probability distribution over the next token.

All training is performed using the Adam optimizer, with a learning rate of $10^{-4}$. Supervised training used a batch size of 128 and RL methods used a batch size of 16. We used 100 rollouts per samples

| Hero facing North |
| Hero facing South |
| Hero facing West |
| Hero facing East |
| Obstacle |
| Grid boundary |
| 1 marker |
| 2 marker |
| 3 marker |
| 4 marker |
| 5 marker |
| 6 marker |
| 7 marker |
| 8 marker |
| 9 marker |
| 10 marker |

Table 5: Representation of the grid

| | Input Grid | Output Grid |
|---|---|---|
| Grid Embedding | Conv2D, kernel size = 3, padding = 1, 16 → 32 
 ReLU | Conv2D, kernel size = 3, padding = 1, 16 -> 32 
 ReLU |
| Residual Block 1 | Conv2D, kernel size = 3, padding 1, 64 → 64 
 ReLU 
 Conv2D, kernel size = 3, padding 1, 64 → 64 
 ReLU 
 Conv2D, kernel size = 3, padding 1, 64 → 64 
 ReLU | |
| Residual Block 1 | Conv2D, kernel size = 3, padding 1, 64 → 64 
 ReLU 
 Conv2D, kernel size = 3, padding 1, 64 → 64 
 ReLU 
 Conv2D, kernel size = 3, padding 1, 64 → 64 
 ReLU | |
| Fully Connected | Linear, 20736 → 512 | |

Table 6: Encoding of the Input/Output Pairs

for the Reinforce method and a beam size of 64 for methods based on the beam search. The value of C used for the methods computing a loss on bags of programs was 5.

