# OpenReview forum: "Leveraging Grammar and Reinforcement Learning for Neural Program Synthesis"
_ICLR.cc/2018/Conference — Accept (Poster)_

### Official Review · AnonReviewer2 · 2017-11-28
**Good paper, could be more clearly written.**

**Rating:** 5
**Confidence:** 3

**Review:**

The authors consider the task of program synthesis in the Karel DSL. Their innovations are to use reinforcement learning to guide sequential generation of tokes towards a high reward output, incorporate syntax checking into the synthesis procedure to prune syntactically invalid programs. Finally they learn a model that predicts correctness of syntax in absence of a syntax checker.

While the results in this paper look good, I found many aspects of the exposition difficult to follow. In section 4, the authors define objectives, but do not clearly describe how these objectives are optimized, instead relying on the read to infer from context how REINFORCE and beam search are applied. I was not able to understand whether syntactic corrected is enforce by way of the reward introduced in section 4, or by way of the conditioning introduced in section 5.1. Discussion of the experimental results coould similarly be clearer. The best method very clearly depends on the taks and the amount of available data, but I found it difficult to extract an intuition for which method works best in which setting and why.

On the whole this seems like a promising paper. That said, I think the authors would need to convincingly address issues of clarity in order for this to appear.

Specific comments

- Figure 2 is too small

- Equation 8 is confusing in that it defines a Monte Carlo estimate of the expected reward, rather than an estimator of the gradient of the expected reward (which is what REINFORCE is).

- It is not clear the how beam search is carried out. In equation (10) there appear to be two problems. The first is that the index i appears twice (once in i=1..N and once in i \in 1..C), the second is that λ_r refers to an index that does not appear. More generally, beam search is normally an algorithm where at each search depth, the set of candidate paths is pruned according to some heuristic. What is the heuristic here? Is syntax checking used at each step of token generation, or something along these lines?

- What is the value of the learned syntax in section 5.2? Presumaly we need a large corpus of syntax-checked training examples to learn this model, which means that, in practice, we still need to have a syntax-checker available, do we not?

---

> ### Author Response · Authors · 2017-12-27
> **Reply to Reviewer 2**
>
> We thank the reviewer for the detailed comments on how to improve the exposition of our paper, which we included in the revised version.
>
> - Figure 2’s size was increased to make the model clearer.
>
> - Reinforce vs. Monte Carlo estimate of the expected reward:
> Equation 8 indeed describes how we estimate the expected reward, we also added the form of the estimator of the expected gradient for a sample i to make what me meant clearer
>
> Beam search and Approximate probabilities
> Equation 10 indeed had a typo. The i in “i \in 1..C” should have been a “r”, making the product a product of the probability of the C programs sampled from the approximate probability distribution. At a search depth of d, the heuristic used to prune candidate paths is the probability of the prefix (which you can think of as the product of equation (5) but limited to the first d terms).
>
> We arrive at a search depth d with a set of S candidates. For each of these S candidates, we obtain the probability of the next token using the softmax. Combining the probability of this token with the product of the whole path that comes before it, we obtain the probability for S * (nb_possible_token) possible paths. We only keep the S best ones (possibly removing the ones that have reached a termination symbol) and repeat the step at the depth d+1.
> We end up at the end with a set of S samples which are going to be used as the basis for our approximate distribution. We have added more description of the process to make it clearer.
>
> When syntax checking is available, whether in its learned form or not, it is implicitly included as its contribution is introduced just before the softmax (see Figure 2 if you can zoom in). A token judged non-syntactically correct would have a probability of zero, so the probability of the path containing it would be zero and would therefore not be included into the promising paths going to the next stage.
>
>
> - Is there value in learning syntax?
> It might be possible to have access to a large amount of programs in a language without having access to a syntax checker, such as for example if we have downloaded a large amount of programs from a code repository. Moreover, it might be useful even for common languages: Note that what we require is a bit different to a traditional syntax checker: answering the question “is this program syntactically correct”, which any compiler would give; as opposed to what we have in equation 13 which corresponds to “Do these first t tokens contain no syntax error and may therefore be a valid prefix to a program”. The syntax checker we need has to return a decision even for non-complete program, therefore it would require some work to transform current compilers to return such answers.
> Finally, as shown in our experiments, using a learned syntax checker might perform better than using a formal one, as it can capture what represents an “idiomatic” program vs. a technically correct one.

---

### Official Review · AnonReviewer3 · 2017-11-28

**Rating:** 6
**Confidence:** 3

**Review:**

The paper presents a reinforcement learning-based approach for program synthesis. The proposed approach claims two advantages over a baseline maximum likelihood estimation-based approach. MLE-based methods penalize syntactically different but semantically equivalent programs. Further, typical program synthesis approaches don't explicitly learn to produce correct syntax. The proposed approach uses a syntax-checker to limit the next-token distribution to syntactically-valid tokens.

The approach, and its constituent contributions, i.e. of using RL for program synthesis, and limiting to syntactically valid programs, are novel. Although both the contributions are fairly obvious, there is of course merit in empirically validating these ideas.

The paper presents comparisons with baseline methods. The improvements over the baseline methods is small but substantial, and enough experimental details are provided to reproduce the results.  However, there is no comparison with other approaches in the literature. The authors claim to improve the state-of-the-art, but fail to mention and compare with the state-of-the-art, such as [1]. I do find it hard to trust papers which do not compare with results from other papers.

Pros:
1. Well-written paper, with clear contributions.
2. Good empirical evaluation with ablations.

Cons:
1. No SOTA comparison.
2. Only one task / No real-world task, such as Excel Flashfill.

[1]: "Neural Program Meta-Induction", Jacob Devlin, Rudy Bunel, Rishabh Singh, Matthew Hausknecht, Pushmeet Kohli

---

> ### Author Response · Authors · 2017-12-27
> **Reply to Reviewer 3**
>
> We thank the reviewer for the comments on the paper, and for pointing out the missing related work.
>
> It is difficult to perform an exact comparison between the two papers as they are solving two different problems. The model developed in [1] (Devlin et al, 2017) performs program induction: i.e. it produces the output world on a new input world where the desired program semantics are encoded in the network itself. On the other hand, in our case, we perform program synthesis, i.e. generate a program in the Karel DSL that performs the desired transformation from input to output.
>
> Using the terminology of Devlin et al., what we describe is closest to the meta-induction approach: strong cross task knowledge sharing but no task specific learning. Overall, our MLE baseline architecture will correspond to the Devlin et al. meta-induction architecture if the decoder was trained to generate program tokens instead of output worlds. This precision was added to the paper
>
> -> No real-world task such as FlashFill?
> The FlashFill DSL considered in previous neural program synthesis work such as RobustFill is essentially a functional language comprising of compositions of a sequence of functions. In this work, we wanted to increase the complexity of the DSL one step further to better understand what neural architectures are more appropriate for learning programs with such complexity. Concretely, the Karel DSL consists of complex control-flow such as nested loops and conditionals, which are not present in the FlashFill DSL. The difference of performance of meta-induction on FlashFill (~70% from  Figure 7 of [2]) vs. KarelDSL (~40% from Figure 4 of [1]) points towards Karel being a more complex dataset.
>
>  Learning Karel programs can also be considered close to a real-world task as this language is used to teach introductory programming to Stanford students, and the program synthesis models can be used to help students if they are having difficulty in writing correct programs.
>
>
> [1] Jacob Devlin, Rudy Bunel, Rishabh Singh, Matthew Hausknecht, Pushmeet Kohli. Neural Program Meta-Induction. In NIPS, 2017
> [2] Jacob Devlin, Jonathan Uesato, Surya Bhupatiraju, Rishabh Singh, Abdel-rahman Mohamed, and Pushmeet Kohli. Robustfill: Neural program learning under noisy I/O. In ICML, 2017

---

### Official Review · AnonReviewer1 · 2017-12-01
**Good paper, accept**

**Rating:** 7
**Confidence:** 3

**Review:**

This is a nice paper. It makes novel contributions to neural program synthesis by (a) using RL to tune neural program synthesizers such that they can generate a wider variety of correct programs and (b) using a syntax checker (or a learned approximation thereof) to prevent the synthesizer from outputting any syntactically-invalid programs, thus pruning the search space. In experiments, the proposed method synthesizes correct Karel programs (non-trivial programs involving loops and conditionals) more frequently than synthesizers trained using only maximum likelihood supervised training.

I have a few minor questions and requests for clarification, but overall the paper presents strong results and, I believe, should be accepted.


Specific comments/questions follow:


Figure 2 is too small. It would be much more helpful (and easier to read) if it were enlarged to take the full page width.

Page 7: "In the supervised setting..." This suggests that the syntaxLSTM can be trained without supervision in the form of known valid programs, a possibility which might not have occurred to me without this little aside. If that is indeed the case, that's a surprising and interesting result that deserves having more attention called to it (I appreciated the analysis in the results section to this effect, but you could call attention to this sooner, here on page 7).

Is the "Karel DSL" in your experiments the full Karel language, or a subset designed for the paper?

For the versions of the model that use beam search, what beam width was used? Do the results reported in e.g. Table 1 change as a function of beam width, and if so, how?

---

> ### Author Response · Authors · 2017-12-27
> **Reply to Reviewer 1**
>
> We thank the reviewer for the comments on the part of the paper that need clarification. We incorporated his feedback in the new version. Here are answers to the raised questions:
>
> -> Figure 2 is too small
> The size of Figure 2 was increased to make it full page width.
>
> -> “The syntaxLSTM can be trained without supervision in the form of known valid programs”
> While the syntaxLSTM can be trained without access to known valid programs when RL-based training is employed (because gradient will flow to it through the softmax defining the probability of each token), we point out in the experiments section that we weren’t successful in training models using only RL training. As a result, it would not be accurate to claim that it can be trained without any valid programs as supervision.
>
> -> Is the Karel DSL the full Karel language?
> The exact description of our DSL is in Appendix B. It doesn’t exactly match the full Karel language, as most notably there is no possibility to define subroutines. We made this clearer in the paper.
>
> -> What was the beam width used?
> All of our experiments used a beam width of 64. We didn’t study the effect of this hyperparameter and chose it as the maximum width we could afford based on available GPU memory. In the limit, using an extremely large beam size would be equivalent to computing the complete sum for the expected reward but this is not feasible for any applications where program would be longer than a few tokens.

---

### Public Comment · ~Taehoon_Kim1 · 2017-12-21
**Questions about reproducibility of the paper**

Dear authors,

Thanks for your interesting paper that I enjoyed a lot. It is great to read new approaches in the field of program synthesis and its promising results. I believe the contributions of the paper are clear but the experimental details are not sufficient to reproduce the results. Below is the list that I found missing in the paper:

1. Sampling method for input/output grid world (ex. # of markers, # of obstacles, Code blocks like repeat(19) { repeat (15) { ... }} or repeat(17) { turnRight } might work as noises)
2. Sampling method for Karel program (ex. max # of tokens, max depth of program)
3. How to deal with corner cases like program with endless loop
4. 52 tokens for Karel DSL
5. Batch size
6. Detailed on beam search

Because the sampling methods of world and program are critical to set the difficulty of the problem, I think the authors could discuss in more details about it to extend the suggested methods. Can the author offer some details on this?

The current attempt to reproduce the Karel dataset can be found https://github.com/carpedm20/karel and https://github.com/carpedm20/program-synthesis-rl-tensorflow.

---

> ### Author Response · Authors · 2017-12-27
> **Reply to Commenter**
>
> We thank the commenter for their interest in our paper.
>
> More details about the generation of the dataset are available in a previous paper that was making use of the Karel Dataset [1]. We have already released the karel dataset (link omitted because of double-blind constraints) and also plan on releasing the code used to run the experiments. What follows are the answers to the specific questions:
>
> 1, 2, 3 -> The input grids are generated by randomly sampling for each cell whether the cell contains an obstacle / a marker / several markers. The agent’s position is also selected at random.
> A program is then sampled with a maximum nesting depth (nested loops or conditionals) of 4 and a maximum number of tokens that is set to be 20. We execute the programs on the inputs grids to generate the output grids. If the program hasn’t halted before performing 200 actions, if there is a collision with an obstacle or if the program doesn’t do anything when run on the sampled grid (i.e. the input grid is unchanged), we discard the program and sample a new one.
>
> 4 -> The 52 tokens are: <s> (start of sequence), not, DEF, run, REPEAT, WHILE, IF,  IFELSE,  ELSE, markersPresent, noMarkersPresent, leftIsClear, rightIsClear, frontIsClear, move, turnLeft, turnRight, pickMarker, putMarker, m(, m) (open and close parens for a function), c(, c) (open and close parens for a conditional),  r(, r) (open and close parens for a repeat instruction), w(, w) (open and close parens for a while conditional), i(,i) (open and close parens for a if statement conditional), e(, e) (open and close parens for an else clause) + 20 scalar values from 0 to 19.
>
> 5 -> The batch size used for the supervised setting was 128. For the RL type experiments, a batch was composed of 16 samples. We used 100 rollouts per samples for the Reinforce method and a beam size of 64 for methods based on the beam search.
>
> 6-> We have added more details in the paper regarding how the beam search is performed.
>
> [1] Jacob Devlin, Rudy Bunel, Rishabh Singh, Matthew Hausknecht, Pushmeet Kohli. Neural Program Meta-Induction. In NIPS, 2017

---

> > ### Public Comment · (anonymous) · 2018-01-30
> > **Further questions about data**
> >
> > Thank you for the additional clarifications about the dataset. I have some further questions that I am hoping you will be able to answer.
> >
> > > We have already released the karel dataset (link omitted because of double-blind constraints)
> > Now that the double-blind period is over, can you confirm whether what you used for this paper is the same dataset as http://bit.ly/karel-dataset?
> >
> > In your reply above, you state
> > > The input grids are generated by randomly sampling... A program is then sampled
> > However, the current version of the paper states (on page 8, section 6.1)
> > > randomly sampling programs from the DSL... or each program, we a set of IO pairs are generated by sampling random input grids
> > which seems to be in the opposite order. Could you clarify which of these two procedures was followed?
> >
> > > maximum nesting depth (nested loops or conditionals) of 4 and a maximum number of tokens that is set to be 20
> > I looked through the train.json file in the dataset from http://bit.ly/karel-dataset, but I found an example which doesn't fit the constraints (line 144701 in the file, GUID 8a4df93b40edfa61):
> > > DEF run m( REPEAT R=5 r( turnLeft WHILE c( rightIsClear c) w( REPEAT R=3 r( REPEAT R=4 r( IFELSE c( not c( frontIsClear c) c) i( REPEAT R=2 r( WHILE c( not c( rightIsClear c) c) w( turnLeft w) r) turnRight i) ELSE e( move e) r) r) putMarker IF c( leftIsClear c) i( turnRight i) w) r) turnLeft m)
> > This is 60 tokens long and has a maximum nesting depth of 7 (REPEAT R=5 -> WHILE rightIsClear -> REPEAT R=3 -> REPEAT R=4 -> IFELSE not frontIsClear -> REPEAT R=2 -> WHILE not rightIsClear).
> >
> > For this paper, did you use a different version of the dataset which contains simpler programs than this example from http://bit.ly/karel-dataset?

---

> > > ### Author Response · Authors · 2018-02-15
> > > **Answers to Further questions**
> > >
> > > Hello, thank you for your interest in this paper.
> > >
> > > 1) Yes, this is the exact dataset (train, test, val) that we use for our experiments. (bit.ly/karel-dataset)
> > >
> > > 2) It doesn't matter in this case if we sample programs first or I/O examples first since they are independently sampled here. Program and Input grid are independently sampled, and output grids are the results of applying the program to the Input grid.
> > >
> > > 3) You are absolutely right. Our previous reply was mistakenly giving you statistics about an earlier version of the dataset which we didn't use for this paper. I'm terribly sorry about this mistake and apologize for the confusion. For this dataset, the programs contain up to 75 tokens and have a maximum nesting depth of 8 (counting the function definition as 1).

---

### Decision · Program_Chairs · 2018-01-29
**ICLR 2018 Conference Acceptance Decision**

**Decision:**

Accept (Poster)

**Comment:**

Below is a summary of the pros and cons of the proposed paper:

Pros:
* Proposes a novel method to tune program synthesizers to generate correct programs and prune search space, leading to better and more efficient synthesis
* Shows small but substantial gains on a standard benchmark

Cons:
* Reviewers and commenters cited a few clarity issues, although these have mostly been resolved
* Lack of empirical comparison with relevant previous work (e.g. Parisotto et al.) makes it hard to determine their relative merit

Overall, this seems to be a solid, well-evaluated contribution and seems to me to warrant a poster presentation.

Also, just a few notes from the area chair to potentially make the final version better:

The proposed method is certainly different from the method of Parisotto et al., but it is attempting to solve the same problem: the lack of consideration of the grammar in neural program synthesis models. The relative merit is stated to be that the proposed method can be used when there is no grammar specification, but the model of Parisotto et al. also learns expansion rules from data, so no explicit grammar specification is necessary (as long as a parser exists, which is presumably necessary to perform the syntax checking that is core to the proposed method). It would have been ideal to see an empirical comparison between the two methods, but this is obviously a lot of work. It would be nice to have the method acknowledged more prominently in the description, perhaps in the introduction, however.

It is nice to see a head-nod to Guu et al.'s work on semantic parsing (as semantic parsing from natural language is also highly relevant). There is obviously a lot of work on generating structured representations from natural lanugage, and the following two might be particularly relevant given their focus on grammar-based formalisms for code synthesis from natural language:

* "A Syntactic Neural Model for General-purpose Code Generation" Yin and Neubig ACL 2017.
* "Abstract Syntax Networks for Code Generation and Semantic Parsing" Rabinovich et al. ACL 2017